# SPARSE REWARDS CAN SELF-TRAIN DIALOGUE AGENTS

## ABSTRACT

Recent advancements in state-of-the-art (SOTA) Large Language Model (LLM) agents, especially in multi-turn dialogue tasks, have been primarily driven by supervised fine-tuning and high-quality human feedback. However, as base LLM models continue to improve, acquiring meaningful human feedback has become increasingly challenging and costly. In certain domains, base LLM agents may eventually exceed human capabilities, making traditional feedback-driven methods impractical. In this paper, we introduce a novel self-improvement paradigm that empowers LLM agents to autonomously enhance their performance without external human feedback. Our method, Juxtaposed Outcomes for Simulation Harvesting (JOSH), is a self-alignment algorithm that leverages a sparse reward simulation environment to extract ideal behaviors and further train the LLM on its own outputs. We present ToolWOZ, a sparse reward tool-calling simulation environment derived from MultiWOZ. We demonstrate that models trained with JOSH, both small and frontier, significantly improve tool-based interactions while preserving general model capabilities across diverse benchmarks. Our code and data are publicly available on GitHub at `https://anonymous.4open. science/r/josh_iclr-C8DE/README.md`

## 1 INTRODUCTION

Large Language Models (LLMs) (Bommasani et al., 2021; Brown et al., 2020; Achiam et al., 2023) have shown a well-marked ability to follow instructions under various tasks. These advancements are often attributed to post-training fine-tuning based on human preferences. This includes multi-turn tool calling tasks where an LLM-based agent must solve a task by interacting with both a user and a set of tools (or APIs) (Farn & Shin, 2023; Yao et al., 2024a). Further task-specific alignment for tool-calling tasks can take the form of preference judgments. But these can be expensive to obtain. Furthermore, there is usually a more 'crisp' notion of success for such tasks. Specifically, was the right tool(s) or API(s) called with the right set of arguments? Ideally, alignment should be optimized towards these sparse rewards.

In this paper, we propose a self-alignment process JOSH (Juxtaposed Outcomes for Simulation Harvesting) that can be used to improve a model's performance on multi-turn tool calling by optimizing for tool/API completion using simulated rollouts of reward-conditioned conversations. We show that this method is general and can be applied to weak/small or frontier LLMs, though gains are significantly larger for the former. We also present a new tool calling benchmark, ToolWOZ, refashioning MultiWoz2.0 (Zang et al., 2020) to train and evaluate multi-turn tool calling effectively.

JOSH utilizes a beam search inspired simulation approach, employing sparse rewards (in this paper corresponding to successful tool calls) to guide turn-by-turn generation and synthesize preference-annotated examples. JOSH allows an agent to generate multiple responses at each conversation turn, exploring various trajectories through a conversation until a sparse reward (goal) is encountered along some path. Upon reaching a goal, other beam candidates are pruned and further expansion proceeds only from that point. Once a trajectory achieves all possible goals, all remaining trajectories are backtracked, logging unsuccessful paths as negative alignment samples and successful paths as positive ones. This process constructs alignment preference data solely from the model itself. When used to align that same model, we show it enhances model performance.

Tool use is a critical skill in LLMs (Mialon et al., 2023; Schick et al., 2024); however, there is a large disparity in tool-using capabilities across different model sizes, especially when involving them in

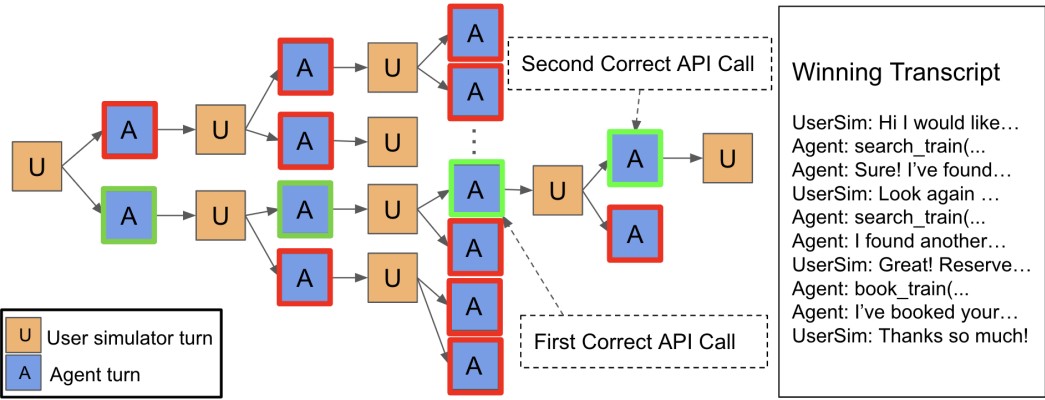

Figure 1: Illustration of JOSH, a tool calling simulation-based beam search experience generation algorithm. A correct path through the conversation can be mapped out (shown in green) by backtracking from sparse rewards achieved by the agent. In this scenario, the sparse rewards are represented by "correct" API calls called by the agent. From backtracking through the tree an "ideal" path through the conversation is found and training data can be extracted.

multi-turn reasoning (Gao et al., 2024). Furthermore, existing benchmarks either lack a real-world multi-turn setup (Ruan et al., 2024) or intentionally keep the agent's dialogue disjoint from underlying databases and focus more on tool selection (Huang et al., 2023). To demonstrate JOSH's capability to improve a model used in an agentic system through self-alignment, we introduce a new dataset ToolWOZ. Derived from the task-oriented dialogue (TOD) benchmark MultiWOZ, ToolWOZ is a multi-turn tool-based simulation environment where an agent model is assessed on its tool-calling capabilities by calling goal APIs through collaboration with a user simulator. After self-alignment using JOSH on the ToolWOZ training set, a `meta-llama3-8B-instruct` (Meta-Llama, 2024) model exhibits a 74% increase in Success Rate. Additionally, after JOSH self-alignment we see `gpt-4o` beats it's own baseline to become state-of-the-art on two separate benchmarks: ToolWOZ and $\tau$-bench (Yao et al., 2024a).

To show that JOSH does not degrade general model performance, we evaluate a trained `meta-llama3-8B-instruct` model across three public benchmarks: MT-Bench (Zheng et al., 2024), $\tau$-bench , and the LMSYS-Chatbot Arena Human Preference Predictions challenge (lin Chiang, 2024; Zheng et al., 2024). Our results confirm that the JOSH aligned model does not regress relative to its base model, despite its new specialized alignment.

## 2   JOSH: JUXTAPOSED OUTCOMES FOR SIMULATION HARVESTING

In this section, we detail the two components of JOSH, our method for generating synthetic preference-annotated training data to enhance tool-driven multi-turn dialogue. We use the terms "tool" and "API" interchangeably. Our approach for generating conversations uses an agent-simulator system and involves a turn-level beam search strategy combined with tool/API-calling reward pruning. Unlike traditional token-level beam search, our method maintains multiple active multi-turn conversations (trajectories) over sequences of agent turns. From these synthetic conversations we create preference-annotated training instances. This involves extracting both supervised fine-tuning (SFT) data and preference fine-tuning (PFT) data. The SFT data is derived from the optimal trajectory (or path) through the conversation tree, while the PFT data includes pairwise comparisons of good and bad agent turns, facilitating more nuanced model training.

### 2.1   BEAM SEARCH SIMULATION FOR COLLECTING USER-AGENT CONVERSATIONS

We begin by having the agent $A$ (defined in Section 4.2) interact with a user simulator $U$ (defined in Section 4.3). A set of goals $G = \{g_1, g_2, \ldots, g_k\}$ is defined where achieving a goal will award the agent $A$ a sparse reward of value $\frac{1}{len(\text{Goal Set})}$ to it's return. Our return uses the Average Reward formulation (Sutton & Barto, 2018) hence we denote it as $AR$ and refer to it as "Average Reward".

---

**Algorithm 1** Beam Search Simulation for Collecting User-Agent Conversations

---

1: Input parameters: max_depth; branching_factor; max_beam
2: Load: $U \leftarrow$ User Simulator; $A \leftarrow$ Agent; $G \leftarrow$ Goal Set
    $* \; U(l)$ and $A(l)$ run one turn of the user and agent on a leaf node $l$ of a conversation trajectory.

3: // Initialize control parameters and data structures
4: $AR \leftarrow 0$    // Average reward
5: $leaves \leftarrow []$ // Trajectory leaf nodes which will be expanded in beam search
6: $depth \leftarrow 0$  // Current trajectory depth

7: **while** depth $\leq$ max_depth **and** $G \neq \emptyset$ **do**
8:    // Expand trajectories by running user simulation and agent.
9:    $leaves = [U(l) \;\; \forall l \in leaves]$  // Expand with next user response by running the simulator one step on all leaves.
10:    // Expand trajectories by running agent $A$ on all leaves.
11:    // Each expansion is a full turn of $A$ including API calls, thoughts and utterances.
12:    **if** len($leaves$) $*$ *branching_factor* $\leq$ *max_beam* **then**
13:        $leaves = [A(l) \;\; \forall l \in leaves]$
14:    **end if**

15:    // Check for goals. If a goal is reached, prune trajectories to retain successful path.
16:    **for** $leaf \in leaves$ **do**
17:        // If a goal was reached in leaf
18:        **if** $\exists g \in G$ **and** $g \in leaf]$ **then**
19:            // Set leaf as the new root, remove $g$ from Goal Set and update the reward.
20:            $leaves = [leaf]$
21:            $G.remove(g)$
22:            $AR = AR + \frac{1}{len(\text{Goal Set})}$
23:            **BREAK**
24:        **end if**
25:    **end for**

26:    $depth \leftarrow depth + 1$
27: **end while**

---

We considered several reward structures for the task of agent dialogues, we found that the cumulative reward method encourages excessive API calls, leading to inefficiency, which is contrary to our aim of minimal interaction for issue resolution. Per-turn rewards, while potentially speeding up learning, necessitate costly annotations or the use of a resource-intensive LLM judge, which we reserve for future exploration. Sparse goal-based rewards, akin to our approach, issue rewards only upon goal completion, offering feedback at each API call to refine agent behavior in real time. While shaped rewards might expedite learning by guiding agents with intermediate incentives, they complicate the reward structure and risk diverting focus from final objectives. By employing an average reward function with partial sparse rewards, we facilitate efficient task completion without the complexities of other structures, ensuring goal-oriented and concise dialogues.

We begin with $AR = 0$. Goals in $G$ can be achieved when $A$ interacts with $U$ in a desired manner. In this paper, rewards are granted when the agent successfully makes a predefined correct tool or API call during a conversation. Figure 1 illustrates an example where the goal set $G$ is composed of multiple correct API calls made within a simulated conversation.

The beam search simulation, in which agent $A$ and user simulator $U$ interact, is detailed in Algorithm 1. The algorithm begins by constructing a tree: the user simulator $U$ initiates the conversation, and agent $A$ generates *branching_factor* agent responses with a high temperature to encourage variability. Each agent turn $A(l)$ – where $l$ is the leaf node of a conversation trajectory – represents a full response, during which the agent may make API calls or other actions before replying to the user.

Following each agent turn, $U$ generates a response, after which $A$ generates another set of *branching_factor* turns for each response from $U$. This continues until an agent turn achieves a goal $g$. In the case of Figure 1, the goal is the "First Correct API Call." The agent turn that achieves this goal becomes the new root, $g$ is removed from the goal set $G$, the Average Reward is increased by $\frac{1}{len(\text{Goal Set})}$, and the process repeats. The goal $g$ is removed from the goal set in order to prevent rewards for duplicate goals. If another turn simultaneously achieves the goal $g$, it is considered partial credit: it does not follow the ideal path but is not considered a negative example either. When the number of branches reaches the *max_beam* size, only one agent response is generated. This process continues until all the goals in the goal set $G$ are achieved or a maximum number of turns is reached. Because the beam search is designed to follow paths once goals are hit, this naturally selects for trajectories where goals are achieved earlier in the conversation.

In this paper, we branch at the turn level rather than the agent action level, allowing the tree to grow exponentially with the number of turns rather than individual actions (i.e., utterance, thoughts, API/tools). Binary trees have a number of $2^{h-1}$ leaf nodes where h is the height of the tree, since JOSH splits at the turn level we can expect $t = \log_2(max\_beam) + 1$ to be the number of turns $t$ before JOSH can no longer expand the tree. There are roughly 3 actions a per turn on average, so the number of branching turns allowed when when action splitting is $t = \frac{\log_2(max\_beam)+1}{3}$. Thus when $max\_beam = 8$ which is used throughout the paper to keep costs reasonable (around \$100) we could perform either $t = 4$ turns while turn splitting, or $t = \frac{4}{3}$ turns when splitting on actions. While splitting on actions may provide more diversity, over the course of a multi turn dialogue we can explore more possible paths deep in the tree for the same $max\_beam$ when splitting on turns.

## 2.2 Preference Data Extraction

Once Algorithm 1 terminates, we have a tree that resembles Figure 1, from which we can extract training data for both Supervised Fine-Tuning (SFT) and Preference Fine-Tuning (PFT).

For SFT, training data is created by backtracking up the tree from the final successful turn to the initial user-simulated utterance. We refer to this as the "ideal path," illustrated by following the green agent turns up the tree in Figure 1, starting from the "Second Correct API Call." This ideal path corresponds to the best agent turns generated to maximize the number of rewards achieved. This data can subsequently be used to train models, guiding them to produce responses that are likely to yield higher rewards. This approach is similar to offline RL with Decision Transformers, where an optimal path is found by conditioning on the reward (Chen et al., 2021).

For PFT, we use the same tree but additionally take advantage of suboptimal model outputs. We create pairwise data by backtracking through the tree in the same manner as for SFT data extraction. At each user turn along the ideal path, we create a (good, bad) agent turn pair. The "good" agent turn is on the ideal path (green in Figure 1), and the "bad" is the alternative agent turn from that user utterance. If the alternative agent turn also leads to a reward but is ultimately not part of the ideal path, it is not used as a negative example. This paper focuses on using pairwise turns, so agent turns that do not stem from a user turn on the ideal path are not included in the training data.

Preference tuning approaches, such as DPO (Rafailov et al., 2024), require pairwise model generations. However, since Algorithm 1 creates pairwise turns where the paired turns can contain different numbers of model generations (e.g., an API call and an agent response), we focus on a more flexible training approach, KTO (Ethayarajh et al., 2024). KTO works by assigning "upvotes" to good examples and "downvotes" to bad examples. Thus, we can still extract pairwise data at the turn level by labeling all agent generations within good turns along the ideal path as upvotes and the alternative turns as downvotes, without needing model generations to necessarily share the same context. For easy reference, we suffix SFT and KTO preference-tuned models by `-SFT` and `-KTO`, respectively.

## 3 ToolWOZ

In this section, we introduce ToolWOZ, a dataset designed to train and benchmark an agent's capabilities in using API tools within a dialogue setting. ToolWOZ features an agent with access to various API tools that allow it to interact with a real database, engaging in dialogue with a user simulator. Each conversation in the dataset is grounded in seed goals, which correspond to specific goal API calls. As illustrated in Figure 2, ToolWOZ significantly simplifies the analysis of task-oriented dialogue systems compared to earlier DST systems and the MultiWOZ database.

### 3.1 Tool-Calling Datasets for End-to-End Dialogue Systems

In recent years, task-oriented dialogue systems were typically developed using a pipeline approach (Ohashi & Higashinaka, 2022; Mrkšić et al., 2016; Zhang et al., 2020). These systems were divided into multiple components where each component was often modeled with a separate machine learning or natural language processing model, and the datasets used to build these systems, such as MultiWOZ, were designed accordingly.

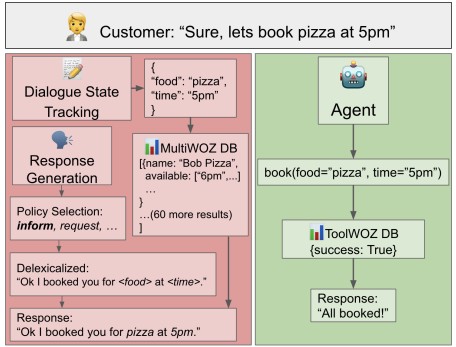

Figure 2: MultiWOZ+DST (left) vs. ToolWOZ+Agent (right) paradigmns for Task Oriented Dialogue interactions.

| Split | # of Data Points |
|---|---|
| Train | 6251 |
| Val | 793 |
| Test | 805 |
| (Official) Test | 450 |
| single-domain dialogues | 2439 |
| multi-domain dialogues | 5410 |

Table 1: ToolWOZ dataset split sizes. This paper uses the first 450 conversation in the ToolWOZ test set as the official test set.

However, with the advent of large language models (LLMs), we are witnessing a shift towards more powerful and reliable end-to-end dialogue systems (Wu et al., 2023a;b), making existing dialogue datasets for pipeline based approaches no longer sufficient for developing models. Recent research has emphasized improving and assessing tool-calling capabilities in dialogue systems, which has become a critical proxy for task-solving and goal achievement. We propose transforming MultiWOZ into a tool-calling benchmark, which can drive the development and evaluation of modern dialogue systems in the LLM era. Moreover, this approach can be generalized to other existing dialogue datasets, enabling a more cost-effective way to create benchmarks for next-generation dialogue systems.

## 3.2 CREATING TOOLWOZ

The design of ToolWOZ addresses several limitations commonly observed in traditional dialogue datasets. One of the key improvements is a shift from indirect metrics like Inform and Success rates to a more direct one, correct API call metric, which measures whether the system can successfully invoke the appropriate external tools (e.g., APIs) based on user inputs. Furthermore, the framework introduces a seamless integration between dialogue and external databases, which helps avoid inconsistencies between the model's actions and the database outcomes. This is complemented by the use of a flexible, goal-oriented user simulator, which allows for repeatable and adaptive interactions with the TOD model. ToolWOZ stands out in Table 1 as a large, domain-diverse multi-turn dialogue benchmark grounded in real-world APIs, containing 7,849 scenarios across various task-based domains.

To create ToolWOZ, we developed APIs for the find (which we refer to as search) and book intents within each of the four MultiWOZ domains that have databases: restaurant, hotel, train, and attraction. Notably, the attraction domain does not include a booking intent. This process yielded the following set of possible APIs: {*search_restaurant*, *book_restaurant*, *search_hotel*, *book_hotel*, *search_train*, *book_train*, and *search_attraction*}. The arguments for each API correspond to the possible slot values for each domain's intent, and all arguments are optional. For full API definitions, refer to the Appendix C. Every ToolWOZ conversation contains a list of goal API calls. A goal API $g(x)$ is considered completed if an agent called function $f(y)$ where $x \subseteq y$ or $g(x)$ and $f(y)$ return only one result from the database, which is the same. Thus, for each conversation, we can quantify its success by the percentage of goal APIs that were called by the agent. This provides a far less gameable notion of correctness as opposed to Inform and Success rate which rely on fuzzy matching of goal states. Goal APIs in ToolWOZ have a loose ordering to them, an agent must generally make a correct search call in order to obtain the necessary information to make a booking in that intent. This simulates real scenarios where agents often need to condition on information from earlier tool calls to make new ones. Goals can, however, be achieved in any order.

ToolWOZ aligns the dialogue and database by only returns correct results when they closely match a goal api. This system ensure that failed searches or booking accurately return either an incorrect result or no result at all. The search and booking algorithms, as well as the rules for cleaning database results, are detailed in Appendix B. Every MultiWOZ conversation also has a list of user goals. We use the user goals to create a goal-oriented user simulator that tries to accomplish the listed goals in order while conversing with the TOD agent. See Section 4.3.

# 4 EXPERIMENTS

## 4.1 DATA AND METRICS

We evaluate different systems on ToolWOZ and $\tau$-bench (Yao et al., 2024a). Similar to ToolWOZ, $\tau$-bench is a recently introduced tool-based multi-turn dialogue LLM benchmark. We only adopt data from the Retail domain in $\tau$-bench, as it contains both training and test data (Airline domain only contains test data).

We use Average Reward and 100% API Success Rate as the two key elements of evaluation to compare models over ToolWOZ. Following (Yao et al., 2024a), we report Pass^1 on $\tau$-bench, which uses final database states to measure the binary success of a conversation. Note that the metric is very similar to the 100% API Success Rate, and the major difference is that 100% API Success Rate considers both Read and Write API calls, while Pass^1 only considers Write APIs. We run $\tau$-bench results 10 times and take the final Pass^1 score to reduce variance, as discussed in Section 5.1.

## 4.2 AGENTS

We benchmarked `gpt-4o-mini` and `gpt-4o` on both ToolWOZ and $\tau$-bench. We also evaluated `gpt-3.5` and `meta-llama-3-8B` (AI@Meta, 2024) on ToolWOZ. For `gpt` models, we explored both Function Calling (Schick et al., 2024) (FC) and ACT/ReACT (Yao et al., 2022) techniques, while for `meta-llama-3-8B`, we used ReACT for all experiments. The prompt used for ReACT models on ToolWOZ is detailed in Appendix Table D.

Using goal API calls as sparse rewards, we generated JOSH rollouts for models on ToolWOZ across 926 conversations in the ToolWOZ training set. For models on $\tau$-bench, we generated JOSH rollouts for all 500 conversations in the Retail domain training set.

On both ToolWOZ and $\tau$-bench, the JOSH rollout process involved a max beam size of 8 and a branching factor of 2. We do experimentation in section 4.4 to explore other beam sizes. For `meta-llama-3-8B`, sampling parameters were set at temperature 1.5, top_k=50, and top_p=0.75 to foster diverse generation. For `gpt` versions, the temperature was set to 1.0. The average cost of running JOSH on a `meta-llama-3-8B` agent was approximately $0.11 per ToolWOZ conversation, amounting to roughly $102 for all 926 conversations. The average cost of finetuning `gpt-4o` on ToolWOZ was between $75 and $200 depending on the prompting approach. For training all models, we retained only those conversations whose JOSH rollouts achieved 100% success in the ideal path without errors. For `meta-llama-3-8B` on ToolWOZ, this resulted in a final filtered training set of 631 conversations.

From these successful JOSH rollouts, we extracted KTO and SFT data as described in section 2.2. For training `meta-llama-3-8B SFT`, the model was trained for 3 epochs with a learning rate of 2e-5. For `meta-llama-3-8B-KTO`, the model was trained for 1 epoch with a learning rate of 5e-7 and a beta of 0.1. The `meta-llama-3-8B` models were trained using Lora and 4-bit quantization. We fine-tuned `gpt-4o` for 3 epochs, with a batch size of 1, and an LR multiplier of 2.

## 4.3 USER SIMULATORS

We experimented with two types of user simulators, both based on `gpt-4o`: goal-based and guide, to assess their impact on the performance and repeatability of evaluating agents on the ToolWOZ test set. The user simulators were run with a temperature of zero. The goal-based simulator strictly follows the predefined user goals for each conversation, without access to the human-human transcript from the dataset. In contrast, the guide simulator references the MultiWOZ transcript and suggests

| Agent | Avg Reward | 100% Success Rate |
|---|---|---|
| meta-llama-3-8B | 0.63 | 0.34 |
| meta-llama-3-8B-JOSH-SFT | 0.74 | 0.50 |
| meta-llama-3-8B-JOSH-KTO | **0.79** | **0.59** |
| gpt-3.5-ReACT | 0.66 | 0.44 |
| gpt-4o-mini-ReACT | 0.67 | 0.48 |
| gpt-4o-mini-ReACT-JOSH-SFT-beam-4 | 0.85 | 0.72 |
| gpt-4o-mini-ReACT-JOSH-SFT-beam-8 | 0.85 | 0.72 |
| gpt-4o-mini-ReACT-JOSH-SFT-beam-16 | **0.865** | **0.74** |
| gpt-3.5-FC | 0.76 | 0.58 |
| gpt-4o-mini-FC | 0.88 | 0.76 |
| gpt-4o-mini-FC-JOSH-SFT | **0.89** | **0.78** |
| gpt-4o-ReACT | 0.900 | 0.791 |
| gpt-4o-ReACT-JOSH-SFT | **0.914** | **0.813** |
| gpt-4o-FC | 0.919 | 0.831 |
| gpt-4o-FC-JOSH-SFT | **0.922** | **0.84** |

Table 2: ToolWOZ test set results. Those with *-JOSH* in the model name were trained on JOSH rollouts using their base model on the first 926 conversations in the ToolWOZ training dataset.

| Agent | Pass$^1$ |
|---|---|
| gpt-4o-mini-ReACT | 16.87 |
| gpt-4o-mini-ReACT-JOSH-SFT | **36.34** |
| gpt-4o-mini-ACT | 44.60 |
| gpt-4o-mini-ACT-JOSH-SFT | **47.65** |
| gpt-4o-mini-FC | 50.78 |
| gpt-4o-mini-FC-JOSH-SFT | **58.26** |
| gpt-4o-ACT | 63.13 |
| gpt-4o-ACT-JOSH-SFT | **64.26** |
| gpt-4o-ReACT | 54.43 |
| gpt-4o-ReACT-JOSH-SFT | **58.43** |
| gpt-4o-FC | 65.21 |
| gpt-4o-FC-JOSH-SFT | **66.00** |

Table 3: gpt-4o trained on JOSH rollouts on $\tau$-bench Retail. gpt-4o-FC was the previous state-of-the-art on the $\tau$-bench Retail test set (Yao et al., 2024a).

specific quotes from the original dialogue. Detailed prompts for both simulators are provided in Appendix D. While we primarily report results based on the goal-based simulator, a comparative analysis of the two simulators is provided in Section 5. For the $\tau$-bench dataset, we were only able to evaluate the goal-based simulator, as no transcripts are available.

## 4.4 RESULTS

We outline the results of training three models on JOSH rollouts from their respective base models: a smaller meta-llama-3-8B model, gpt-4o-mini, and the larger gpt-4o model. We show that each JOSH trained model variant outperforms their respective baseline variant achieving state-of-the-art performance on both ToolWOZ and $\tau$-bench. Specifically, we show how training on JOSH rollouts makes gpt-4o-FC-JOSH-SFT surpass the vanilla gpt-4o-FC on the $\tau$-bench Retail datasets. Similarly, gpt-4o-FC-JOSH-SFT outperforms the vanilla variant on gpt-4o on ToolWOZ. It is worth noting that JOSH self-alignment can augment gpt-4o ability on tool benchmarks, inspite of gpt-4o already having state-of-the-art ability, being ranked within top 3 on the LM-Sys Chatbot Arena Leaderboard (Chiang et al., 2024) and top 2 on both HELM (Bommasani et al., 2023) and 5-shot MMLU (Hendrycks et al., 2021).

On ToolWOZ, the meta-llama-3-8B-JOSH-KTO model saw a 74% increase in 100% Success Rate and a 25% increase in Average Reward compared to the baseline meta-llama-3-8B model, as shown in Table 2. This jump is noticeably even higher than the meta-llama-3-8B-JOSH-SFT model. The meta-llama-3-8B-JOSH-KTO model even outperforms both gpt-3.5-ReACT and gpt-3.5-FC.

We see likewise see a large performance jump from the baseline gpt-4o-mini-ReACT model to it's JOSH-SFT trained variant, with a 50% increase in 100% Success Rate and a 27% increase in Average Reward. We explore three beam sizes when doing JOSH using gpt-4o-mini-ReACT and note that while a maximum beam size of 16 is marginally better than 8 and 4, we choose to use a beam size of 8 for all other experiments to save cost and efficiency while still taking advantage of a larger beam size. We also observe that the gpt-4o-FC-JOSH-SFT model outperforms its baseline to achieve state-of-the-art results on ToolWOZ. We note that as gpt-4o-FC performs well on ToolWOZ, the headroom for improvement shrinks and thus performance gains from JOSH is smaller than for other baseline models.

Table 3 tells a similar story on the $\tau$-bench Retail test set. Over three different prompting options, ACT, ReACT, and FC, gpt-4o sees significant performance jumps when training on JOSH rollouts. Notably, gpt-4o-mini-ReACT-JOSH-SFT has a 115% increase over it's baseline score. Also, gpt-4o-FC-JOSH-SFT beats its baseline, the previous state-of-the-art model on $\tau$-bench, gpt-4o-FC. This significant jump in performance for each model can be seen after only being trained on JOSH rollouts from their respective baselines on the 500 conversations in the $\tau$-bench Retail training dataset.

## 5 ANALYSIS

### 5.1 AGENT PERFORMANCE STABILITY ACROSS USER SIMULATORS

| Dimension | Human | User Simulator |
|---|---|---|
| naturalness | 4 | 4 |
| conciseness | 3.98 | 3.94 |
| redundant | 3.59 | 3.42 |

(a) Evaluation comparing the goal based user simulator to the ground truth human users from the conversations in MultiWOZ.

| Agent | (Guide) Avg Reward | (Guide) 100% Success Rate |
|---|---|---|
| meta-llama-3-8B | 0.50 | 0.26 |
| meta-llama-3-8B-JOSH-SFT | 0.55 | 0.33 |
| meta-llama-3-8B-JOSH-KTO | **0.59** | **0.38** |

(b) Models trained against goal user simulator on ToolWOZ run against the guide user simulator on the ToolWoz test set.

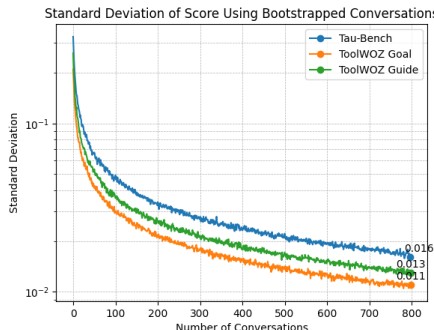

(c) Bootstrap estimation of Standard Deviation of Average Reward on ToolWOZ using two types of user simulators and $\tau$-bench

Figure 3: A deeper look at user simulators and their effects on score stability in benchmarks.

In Figure 3c we examined the stability of the ToolWOZ Average Reward metric across two types of user simulators: goal-based and guide-based. Additionally, we assessed the stability of the $\tau$-bench Pass$^1$ metric by measuring the standard deviation of benchmark scores as the number of conversations increased using the bootstrapping method (Efron, 1992). We observe that all three benchmarks exponentially reduce in standard deviation as the number of samples increases. Notably, the ToolWOZ goal simulator has the lowest standard deviation, which drops below 1.5 percentage points at the 450 samples. Based on this observation, we reduced the ToolWOZ test set to 450 examples, utilizing the goal-based simulator to minimize simulation noise. Additionally, the $\tau$-bench dataset has a high standard deviation of about 4 percentage points at it's test set size of 115. This leads us to run the $\tau$-bench tests 10 times to reduce variability as noted in the previous section.

To evaluate the quality of the goal-based user simulator, we compare it with human users from the ground truth MultiWOZ conversations, as detailed in Table 3a, across three dimensions: naturalness, conciseness, and redundancy. This assessment employs LLM-Rubric Hashemi et al. (2024) prompts using Claude Sonnet 3.5 assigning scores ranging from 1 to 4, with 4 being the highest across all 450 conversations in the ToolWOZ test set. Our findings indicate that both the user simulator and human users score highly on naturalness. However, the user simulator's conciseness is slightly lower than that of human users, attributed to the simulator's tendency towards verbosity. Lastly, the redundancy score for the user simulator is lower compared to human users, primarily due to agent errors prompting the re-request of information. In such cases, the simulator is more inclined to reiterate information, whereas humans are typically less repetitive with critical information.

To ensure robustness and generalization, we further evaluated the JOSH-trained `meta-llama-3-8B` model using rollouts from the goal-based simulator, by testing it against the guide simulator (as described in §4.3). Table 3b demonstrates that the JOSH-trained models consistently outperform the baseline `meta-llama-3-8B` model, regardless of the simulator used. While the distributions of scores vary between the two simulators (as reflected in Table 3b and Table 2), the relative ranking of model performance remains unchanged.

### 5.2 ERROR ANALYSIS

Training on JOSH rollouts additionally led to a large reduction in errors when running the ToolWOZ test set as shown in Table 4a. The `JOSH-KTO` trained model saw a 96% decrease in incorrectly formatted APIs and a 50% reduction in bad API use (e.g. using nonexistant arguments, using nonexistant apis, . . . ). The `JOSH-SFT` model also sees a large drop in error rates in both categories, but similar to the reward measurements it does not perform as well as `JOSH-KTO`.

Furthermore, we see in Figure 4c that in particular search_attraction and search_train have a high disparity in number of errors between SFT and KTO training. To further investigate this phenomenon, we measured the frequency of required argument groups for search_train and search_attraction that the SFT model failed to call.

We observe for search_train that calls with the "arriveBy" argument increases failures from the base model to the SFT model, unlike KTO where the errors drop significantly. We find that this phenomenon is due to the SFT model commonly neglecting to include the "departure" parameter when using the "arriveBy" parameter. The KTO model however avoids this by training on API calls with too few parameters as negative examples, and generally includes all paramters in it's api calls. We observe a similar phenomenon with the search_attraction api, where the SFT model often neglects to use the "type" argument alongside the "area" argument, and also uses invalid "area" arguments such as "area = all". Again the KTO model is able to avoid these pitfalls as many apis with invalid parameters are found in the negative examples.

## 5.3 GENERALIZATION OF JOSH FINE-TUNED MODELS

We evaluated the performance on broader tasks of the `meta-llama-3-8B` models fine-tuned on JOSH rollouts from ToolWOZ across two general-purpose benchmarks—MT-Bench and the LMSYS Chatbot Arena Human Preference Predictions challenge in Table 4b. MT-Bench evaluates chatbots' general knowledge through multi-turn, open-ended questions, while the LMSYS Chatbot Arena Human Preferences challenge measures models' human preference ranking capabilities. For LMSYS, we used the first 1,500 data points as the benchmark.

| Method | Bad API Use | Incorrect API Format |
|---|---|---|
| `meta-llama-3-8B` | 0.40 | 0.25 |
| `meta-llama-3-8B-JOSH-SFT` | 0.24 | 0.09 |
| `meta-llama-3-8B-JOSH-KTO` | 0.20 | 0.01 |

(a) Percentage of conversations with types of API Errors on the ToolWOZ Test Set.

| Model | MT-Bench | LMSYS |
|---|---|---|
| `meta-llama-3-8B` | 7.91 | 0.444 |
| `meta-llama-3-8B-JOSH-SFT` | 7.81 | 0.452 |
| `meta-llama-3-8B-JOSH-KTO` | 7.92 | 0.461 |
| `gpt-4o-FC` | 9.10 | 0.515 |
| `gpt-4o-FC-JOSH-SFT` | 9.12 | 0.514 |

(b) MT-Bench and LMSYS benchmark performance. JOSH rollouts were done on ToolWOZ.

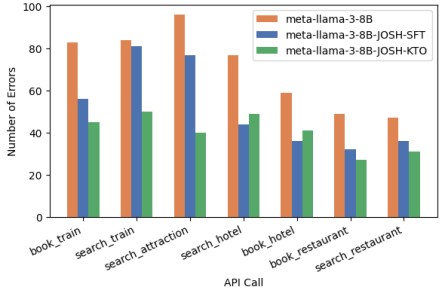

(c) Number of errors caused by ToolWOZ APIs in the Test set

Figure 4: A Further Look at Model Performance - General Benchmarks and Error Analysis

We compared the baseline `meta-llama-3-8B` model, `meta-llama-3-8B-JOSH-SFT`, and `meta-llama-3-8B-JOSH-KTO` on both MT-Bench and LMSYS. As shown in Table 4b, fine-tuning on JOSH rollouts from ToolWOZ did not degrade performance on either benchmark. The models maintained stable performance on multi-domain, multi-turn dialogues (MT-Bench) and human preference ranking (LMSYS).

These results demonstrate that fine-tuning on JOSH rollouts preserves the models' general capabilities. Despite the specific nature of the ToolWOZ training, models adapted to task-oriented dialogue remain effective on broader large language model tasks, with minimal performance degradation.

## 6 RELATED WORK

Notably, simulation environments with sparse rewards were used by DeepMind in their *AlphaGo* (Silver et al., 2016) and *AlphaGo Zero* (Silver et al., 2017) works, enabling the two models to achieve superhuman performance in the game of Go. In the *AlphaGo* works, Monte Carlo Tree Search (MCTS) in a self-play simulation environment is used to intelligently explore the space of next possible moves and long term strategy. Similarly, JOSH treats dialogue as a multi-turn game where it explores possible directions and while using sparse rewards to identify ideal paths through a conversation. JOSH rollouts can then be used to train any LLM for multi-turn tool calling tasks.

With the advent of LLMs (Bommasani et al., 2021; Brown et al., 2020; Achiam et al., 2023) language agents for multi-turn dialogue have seen a sharp increase in effectiveness. Agent reasoning in the dialogue setting has been significantly increased by approaches such as Chain of Thought (COT) (Wei et al., 2022) and ACT/ReACT (Yao et al., 2022) by intelligently scaling inference time compute to reason about a problem before acting. Additionally, dialogue agent's function calling (Schick et al., 2024) abilities have been increased against numerous benchmarks (Patil et al., 2023; Li et al., 2023; Qin et al., 2023b;a). In contrast with ToolWOZ, however, existing tool calling benchmarks lack the proper environment set up for multi-turn dialogue with API goal sets that is suitable for JOSH to run on.

AgentQ (Putta et al., 2024) – a contemporaneous work to this study — is a webagent training and inference process, has similar motivations of self learning using preference data however it has some key differences from JOSH. AgentQ uses MCTS, a self-critique mechanism, and online searching of different pathways, while JOSH is a standalone data extraction algorithm that soley relies on arbitrary sparse rewards. Additionally, AgentQ uses a test time inference strategy while JOSH purely extracts training data for finetuning models, a form of offline RL. JOSH focuses on tool calling multi-turn dialogue while AgentQ is in the domain of navigating web agents. Finally, JOSH training utilizes 100% successful paths to mitigate overfitting on intermediate rewards, while the AgentQ approach requires long horizon exploration to gather preference data.

Other works also explore training LLM agents based on rewards. Approaches such as STaR (Zelikman et al., 2022), Quiet-STaR (Zelikman et al., 2024), and Iterative Reasoning Preference Optimization (Pang et al., 2024) use downstream rewards based on reasoning to train or preference tune models for increased performance at test time. However, these approaches are focused on single turn output performance rather than reasoning in a multi-turn dialogue setting. Some approaches use rewards to train policies to help TOD systems (Hu et al., 2023; Wu et al., 2023b) or extensive test-time search (Snell et al., 2024) while JOSH simply produces data to finetune models rather than make test time changes. In this way JOSH is conceptually similar to Decision Transformers (DTs) (Chen et al., 2021). DTs is a form of offline RL that generates optimal sequences for fine-tuning by conditioning on the rewards, whereas JOSH uses these rewards to select optimal simuation rollouts.

Other approaches use search trees to improve the reasoning of models. Namely Tree of Thought (ToT) (Yao et al., 2024b) and Chain of Preference Optimizaiton (CPO) (Zhang et al., 2024) focus on optimizing the performance of COT reasoning. CPO extracts preference data from a search tree exploring COT reasoning, however it uses an LLM to issue rewards at each sentence and is only applicable to single turn reasoning. On the contrary, JOSH uses sparse rewards in a simulaiton environment to solve reasoning problems in the multi-step dialogue domain.

The preference tuning paradigm was first proposed as an easier to replicate direct supervised learning alternative to well-entrenched "RLHF" paradigm (Ziegler et al., 2019) of first learning a reward model from preferences and then learning a policy using RL-based approaches such as PPO or REINFORCE. Heralded by the first DPO (Rafailov et al., 2024), many variants e.g. RS-DPO (Khaki et al., 2024) and ORPO (Hong et al., 2024) have emerged. Though early approaches required pairwise data with a contrasting good-bad response pair for a prompt, the KTO formulation (Ethayarajh et al., 2024) enabled learning from unpaired preferences. Since the preference data we collect is unpaired, we centrally use KTO in this work.

## 7 CONCLUSIONS

In this work, we devise JOSH, a self-alignment approach which uses sparse rewards to enable agentic models of all sizes to train themselves. We show that training on JOSH rollouts significantly increases performance on benchmarks assessing multi-turn dialogue and tool-calling ability while maintaining or improving general model performance. We show JOSH is general an can be applied to small medium and large models and provide considerable gains in performance. Notably, we illustrate how frontier models can outperform themselves with JOSH to achieve state-of-the-art results on multiple tool-calling benchmarks. Additionally, we present ToolWOZ, a multi-turn, tool-calling simulation dataset with sparse rewards to train and evaluate agent LLMs.

## 8 REPRODUCIBILITY

We have open sourced both the ToolWOZ dataset as well as the JOSH class on GitHub (`https://anonymous.4open.science/r/josh_iclr-C8DE/README.md`). The JOSH class has been designed flexibly, and only requires a step function for an agent and a user in order to begin creating rollouts. The JOSH class also supports a JOSHAgent, JOSHUser, and JOSHRewards base classes to help jump start research and provide an out of the box working solution that can be iterated over. We also provide our implementations for custom JOSH agents on $\tau$-bench. Lastly, we support a parallelized ToolWOZ runner script which allows rapid rollouts of JOSH, fast testing, and supports both local and gpt models.

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

## A  APPENDIX

## B  ALGORITHMS

---

**Algorithm 2** Searching Algorithm

---

1: $args \leftarrow$ API Arguments
2: $d \leftarrow$ Domain
3: $g \leftarrow$ Goals
4: $goal\_parameters \leftarrow g[d]["search"]["parameters"]$
5: $db\_results \leftarrow$ List of served database results
6: $correct\_answer \leftarrow$ None
7: $wrong\_answer \leftarrow$ None
8: $booking\_id \leftarrow$ None
9: // If there is a goal booking call
10: **if** $"book" \in g[d]$ **then**
11:    $booking\_id \leftarrow g[d]["book"]["unique\_id"]$
12: **end if**
13: **for** $result \in db\_results$ **do**
14:    **if** $"book" \in g[d]$ **and** $result["unique\_id"] == booking\_id$ **then**
15:       $correct\_answer \leftarrow result$
16:    **end if**
17:    **if** $goal\_parameters \not\subseteq result$ **then**
18:       $wrong\_answer \leftarrow result$
19:    **end if**
20: **end for**
21: **if** $goal\_parameters \subseteq api\_args$ **then**
22:    **if** $booking\_id$ **then**
23:       **if** $correct\_answer$ **then**
24:          **return** $correct\_answer$
25:       **else**
26:          **if** $wrong\_answer$ **then**
27:             **return** $wrong\_answer$
28:          **else**
29:             **return** $[]$
30:          **end if**
31:       **end if**
32:    **end if**
33: **else if** $args \subseteq goal\_parameters$ **then**
34:    **if** $wrong\_answer$ **then**
35:       **return** $wrong\_answer$
36:    **else if** $booking\_id$ **and** $correct\_answer$ **then**
37:       **return** $correct\_answer$
38:    **end if**
39: **end if**
40: **return** $db\_results[0]$

---

**Algorithm 3** Booking Algorithm

---

1: $args \leftarrow$ API Arguments
2: $d \leftarrow$ Domain
3: $g \leftarrow$ Goals
4: // If there is a goal booking call
5: **if** $"book" \in g[d]$ **then**
6:    **if** $g[d]["book"]["unique\_id"] == args["unique\_id"]$ **then**
7:       $r\_values \leftarrow g[d]["book"]["return"]$
8:       **return** $\{"success" : True, "return" : r\_values\}$
9:    **else**
10:       **return** $\{"success" : False, "return" : None\}$
11:    **end if**
12: **else**
13:    **return** $\{"success" : False, "return" : None\}$
14: **end if**

---

## C TOOLWOZ API SPECS

## D REACT PROMPT

Table 4: API Specifications for ToolWOZ

**API Functions (Part I)**

```
[
  {
    "type": "function",
    "function": {
      "name": "book_restaurant",
      "description": "Allows you to book a restaurant",
      "parameters": {
        "type": "object",
        "required": [],
        "properties": {
          "time": {
            "type": "string",
            "description": "Time the restaurant reservation is at e.g.
                ↪ 13:00"
          },
          "day": {
            "type": "string",
            "description": "Day of the week the restaurant reservation
                ↪ is on e.g. thursday"
          },
          "people": {
            "type": "string",
            "description": "Number of people in the restaurant
                ↪ reservation e.g. 3"
          },
          "name": {
            "type": "string",
            "description": "Name of the restaurant e.g. the river bar
                ↪ steakhouse and grill"
          }
        }
      }
    }
  },
  {
    "type": "function",
    "function": {
      "name": "search_restaurant",
      "description": "Allows you to search a restaurant",
      "parameters": {
        "type": "object",
        "required": [],
        "properties": {
          "food": {
            "type": "string",
            "description": "Type of food served at the restaurant e.g.
                ↪ modern european"
          },
          "pricerange": {
            "type": "string",
            "description": "Price range the restaurant is in e.g.
                ↪ cheap",
            "enum": ["cheap", "expensive", "moderate"]
          },
          "name": {
            "type": "string",
            "description": "Name of the restaurant e.g. jinling noodle
                ↪ bar"
          },
          "area": {
            "type": "string",
            "description": "Area the restaurant is located in e.g.
                ↪ centre"
          }
        }
      }
```

Table 5: API Specifications for ToolWOZ

**API Functions (Part II)**

```
[
  {
    "type": "function",
    "function": {
      "name": "search_hotel",
      "description": "Allows you to search a hotel",
      "parameters": {
        "type": "object",
        "required": [],
        "properties": {
          "name": {
            "type": "string",
            "description": "The name of the hotel e.g. hamilton lodge"
          },
          "area": {
            "type": "string",
            "description": "The area the hotel is located in e.g.
                ↪ north",
            "enum": ["west", "east", "centre", "south", "north"]
          },
          "parking": {
            "type": "string",
            "description": "Whether the hotel offers free parking e.g.
                ↪ yes",
            "enum": ["yes", "no"]
          },
          "pricerange": {
            "type": "string",
            "description": "What the price range of how expensive the
                ↪ hotel is e.g. moderate",
            "enum": ["moderate", "expensive", "cheap"]
          },
          "stars": {
            "type": "string",
            "description": "The number of stars the hotel has e.g. 4",
            "enum": ["0", "1", "2", "3", "4"]
          },
          "internet": {
            "type": "string",
            "description": "Whether or not the hotel has free internet
                ↪ e.g. yes",
            "enum": ["yes", "no"]
          },
          "type": {
            "type": "string",
            "description": "Whether to reserve a hotel or guesthouse.
                ↪ e.g. guesthouse",
            "enum": ["hotel", "guesthouse"]
          }
        }
      }
    }
  }
]
```

Table 6: API Specifications for ToolWOZ

**API Functions (Part III)**

```
[
  {
    "type": "function",
    "function": {
      "name": "search_attraction",
      "description": "Allows you to search an attraction",
      "parameters": {
        "type": "object",
        "required": [],
        "properties": {
          "type": {
            "type": "string",
            "description": "The type or theme of the attraction e.g.
                ↪ boat"
          },
          "name": {
            "type": "string",
            "description": "The name of the attraction e.g. sheep's
                ↪ green and lammas land park fen causeway"
          },
          "area": {
            "type": "string",
            "description": "The area where the attraction is e.g.
                ↪ centre",
            "enum": ["west", "east", "centre", "south", "north"]
          }
        }
      }
    }
  }
]
```

Table 7: API Specifications for ToolWOZ

**API Functions (Part IV)**

```
[
  {
    "type": "function",
    "function": {
      "name": "book_train",
      "description": "Allows you to book a train",
      "parameters": {
        "type": "object",
        "required": [],
        "properties": {
          "people": {
            "type": "string",
            "description": "The number of people or seats to book on
                ↪ the train e.g. 3"
          },
          "trainID": {
            "type": "string",
            "description": "ID for the train the tickets are for e.g.
                ↪ TR2048"
          }
        }
      }
    }
  },
  {
    "type": "function",
    "function": {
      "name": "search_train",
      "description": "Allows you to search a train",
      "parameters": {
        "type": "object",
        "required": [],
        "properties": {
          "leaveAt": {
            "type": "string",
            "description": "Time the train will leave from the
                ↪ departure area e.g. 08:45"
          },
          "destination": {
            "type": "string",
            "description": "Destination area of the train e.g.
                ↪ cambridge"
          },
          "day": {
            "type": "string",
            "description": "Day of the week the train will run e.g.
                ↪ tuesday"
          },
          "arriveBy": {
            "type": "string",
            "description": "Time the train will arrive at the
                ↪ destination e.g. 12:30"
          },
          "departure": {
            "type": "string",
            "description": "Departure area of the train e.g. london
                ↪ liverpool street"
          }
        }
      }
    }
  }
]
```

Table 8: ReACT Prompt for ToolWOZ. Examples are written by hand anecdotally and not taken from the training dataset. Under this setup, the agent will first craft a Plan, then either optionally call an API or SPEAK to the customer. Speaking to the customer ends the agent's turn.

---

You are a customer service agent helping a user.

# General Instructions
You have three commands you can use: PLAN, APICALL, and SPEAK
Always start with a PLAN message, then always end your turn with either a SPEAK or APICALL message.
Your output must include PLAN and APICALL or PLAN and SPEAK.
Each command must be on it's own line. Each line must start with a command.
You must always use commands or else your output will be invalid. Always end your turn with a SPEAK or APICALL message.
Remeber not to use any of the commands unless you are issuing the command.
You MUST finish each command by saying <COMMAND_END>
Remember: After each command, say only <COMMAND_END>

Here is a description of how you should use each command:
## PLAN
Think step by step of what command you will use next and broadly what you should do or say.
Write the plan as an internal thought.
- PLAN should only contain a plan about what you will do. Keep it conscise, the user will never see your plan, instead use SPEAK to communicate with the customer.
- NEVER use PLAN to send a message to the customer.
- You MUST use the apis available to you to gather information. NEVER use your own knowledge, you will be penalized.
- think step by step
- Note: The customer cannot see any PLAN, APICALL, or APIRETURNs
- Be thorough but breif, use logic and reasoning to decide what to do next.
- After recieving an APIRETURN ERROR, write out the API Definition from API Examples in PLAN so you can format the call correctly!
- The SPEAK command ends your turn, so make any APICALLs you need before using SPEAK

## SPEAK
- Always use this command to send a message to the user. This is the only way to talk to the user.
- PLAN will NEVER be sent to the customer.
- Using SPEAK will end your turn, so make any APICALLs you need before using SPEAK

## APICALL
- output the name of the api call you'd like to call. You will have the chance to call more apis if you would like, so call one at a time.
- ONLY output a json dictionary, NEVER output any additional text (example: APICALL {...} <COMMAND_END>)
- Waiting for a response is automatic, NEVER output text relating to waiting for an api response.
- APICALLs and whatever they return are not visible to the customer.
- Use search api calls to search a database and use book api calls to book results from the search.
- NEVER output an api return, it will be given to you after you call APICALL.
- If an APICALL fails, you should try other options. NEVER call the same api more than once, espcially if it didn't work the first time.
- After recieving an APIRETURN ERROR, write out the API Definition from API Examples in PLAN so you can format the call correctly!
- If a parameter is an "enum", those are the ONLY options you can use for that parameter. All other inputs are invalid.

You have the following apis available to you. These are the only apis you have:
### APICALL Specific Instructions
Given a conversation, an api definition, and an example of the api definition filled in, output a valid json dictionary after APICALL and no additional text.

!!! IMPORTANT: You MUST use context clues from the Input to figure out what to assign to each parameter. Never add extra parameters !!!

---

You MUST fill in the parameters based off of the conversation. If a parameter is irrelivant, ALWAYS leave it blank.

### API Definitions
Never add more parameters to the following apis.
HERE ARE THE APICALLs THAT ARE AVAILABLE TO YOU (with example values filled in):
#### API Examples
{example_filled}

Use the conversation to fill in the api definition. You don't have to use all of the parameters if you don't know them. Don't add any new parameters!

If you do not know a parameter, its fine to not include it in the api call.
All parameters are optional.
Note the apicall must be a valid json dictionary with 'name' and 'parameters' keys.
'name' should be a string with the api name. Must be one of the following names ["book_restaurant", "search_restaurant", "book_hotel", "search_hotel", "book_attraction", "search_attraction", "book_train", "search_train"]
'parameters' should also be a valid json dictionary with each key being an argument. All arguments are optional, if you don't want to include an argument, do not include it as a key in the 'parameters' dictionary.

Examples:
## Example 1
Input:
user: Hi I'd like to see if the hamilton loge up north which has parking, 4 stars, internet, moderate pricerange and a guesthouse is available?
Output:
PLAN I will look up a hotel using search_hotel. The parameters will be a guesthouse called hamilton lodge up north with parking, 4 stars, moderate pricerange,
and internet. <COMMAND_END>APICALL {"name": "search_hotel", "parameters": {"name": "hamilton lodge", "area": "north", "parking": "yes", "pricerange": "moderate", "stars": "4", "internet": "yes", "type": "guesthouse"}} <COMMAND_END>

## Example 2
Input:
user: Can I try to book a hotel for this weekend?
Output:
PLAN In order to book the hotel using the 'book_hotel' api, I will need to ask the customer how many people will be coming. <COMMAND_END>SPEAK Sure! How many people will be coming? <COMMAND_END>

## Example 3
Input:
user: Hmmm that date isn't going to work for me, can we try to look for a taxi tomorrow?
Output:
PLAN I do not know what day tomorrow is, I will ask the user. <COMMAND_END>SPEAK Sure! What day is tomorrow? <COMMAND_END>

## Example 4
Input:
user: can you check the trains on tuesday?
assistant: PLAN I'm going to look for trains on tuesday <COMMAND_END>APICALL {"name": "search_train", "parameters": {"leaveAt": "", "destination": "", "day": "tuesday", "arriveBy": "", "departure": ""}} <COMMAND_END>
user: APIRETURN []
Output:
PLAN It seems there are no trains leaving tuesday, I will tell the customer. <COMMAND_END>SPEAK It seems there are no trains leaving tuesday, does any other day work for you? <COMMAND_END>

## Example 5
Input:
user: so I want a train leaving at 8am

assistant: PLAN I will find a train leaving at 8am by using the search_train api call <COMMAND_END>APICALL {"name": "search_train", "parameters": {"leaveAt": "08:00", "destination": "", "day": "", "arriveBy": "", "departure": ""}} <COMMAND_END>

user: APIRETURN []

assistant: PLAN There are no trains that leave at 8am, I will tell the customer. <COMMAND_END>SPEAK Sorry, but there are no trains leaving at 8am. Are there any other times you'd like to look for? <COMMAND_END>

user: Sure how about 9am?

Output:

PLAN I will look up a train using the search_hotel api. The parameters will be parking and high pricerance. <COMMAND_END>APICALL {"name": "search_train", "parameters": {"leaveAt": "09:00", "destination": "", "day": "", "arriveBy": "", "departure": ""}} <COMMAND_END>

## Example 6

Input:

user: so I want a train leaving at 8am with 4 stars

assistant: PLAN I will find a train leaving at 8am with 4 stars by using the search_train api call <COMMAND_END>APICALL {"name": "search_train", "parameters": {"leaveAt": "08:00", "destination": "", "day": "", "arriveBy": "", "departure": "", "stars": "4"}} <COMMAND_END>

user: APIRETURN ERROR

Output:

PLAN The api failed, I will double check the api definition by writing it out. definition: {"name": "search_train", "parameters": {"leaveAt": "", "destination": "", "day": "", "arriveBy": "", "departure":""}. I used a "stars" parameter which is invalid. Let me redo the api with the correct parameters <COMMAND_END>APICALL {"name": "search_train", "parameters": {"leaveAt": "08:00", "destination": "", "day": "", "arriveBy": "", "departure": ""}} <COMMAND_END>

## Example 7

Input:

user: I want a restaurant in the north

assistant: PLAN I will search for a restaurant in the north area using the search_restaurant api call <COMMAND_END>APICALL {"name": "search_restaurant", "parameters": {"food": "", "pricerange": "", "name": "", "area": "north"}} <COMMAND_END>

user: APIRETURN [{...}, {...}, ...]

assistant: PLAN There are multiple options returned, I will inform the customer and ask them if theres any other constraints they are looking for. <COMMAND_END>SPEAK Ok, I found a lot of restaurants in the north, is there anything else you want? <COMMAND_END>

user: Yes I also want it to have room service

Output:

PLAN I will look up a hotel using the search_restaurant api. The parameters are north area and room service. <COMMAND_END>APICALL {"name": "search_restaurant", "parameters": {"food": "", "pricerange": "", "name": "", "area": "north"}} <COMMAND_END>

## Example 8

Input:

user: Wow what a pretty day!

Output:

PLAN The user commented on what a pretty day it is. I will reply that I agree. <COMMAND_END>SPEAK You're right, it is so pretty! <COMMAND_END>

## Final Output

Input:

---

Table 9: Goal based user simulator prompt

---

**SYSTEM:**

You're a customer talking to a travel agent.

You have the following goals you want to accomplish in the conversation (don't relay them all at once to the agent):

{goals}

Discuss with the agent to try and accomplish each one of your goals in order.

If the agent fails at an action, check other goals for a backup plan

Relay information piecemeal to the agent to encourage conversation.

EXCEPTION: Make sure you've communicated all the neccessary information for that intent before proceeding with a booking.

ALL of your listed goals must be fufilled in order for you to agree to a booking.

DO NOT say  or  to the agent.

When you want to end the conversation say END_CONVERSATION

Always say END_CONVERSATION to hang up!

**USER:**

REMEMBER: You are a customer talking to a travel agent.

When you want to end the conversation say END_CONVERSATION

Always say END_CONVERSATION to hang up!

Try to address your next goal or finish the current goal you're focusing on.

Note: if you are looking for a "place to stay", don't refer to it as a hotel unless the goals explicitly state you are looking for a type hotel.

Don't relay all the information about your goal to the agent at once.

ABOVE ALL ELSE, it is critical ALL of your listed goals are fufilled in order for you to agree to a booking. Double check each of your requirements and tell the agent if one is not met. If you're not sure, double check.

EXCEPTION: Make sure you've communicated all the neccessary information for that intent before proceeding with a booking.

If the agent fails at an action, check other goals for a backup plan.

Remeber, you are the customer.

CUSTOMER:

Table 10: Guide based user simulator prompt

**USER:**

You are a coach giving tips to a user simulator trying to replicate a conversation as consistently as possible. The user simulator is in the middle of a conversation, give it advice on what to do in the next turn.

Consistency means that over multiple runs, the user simulator should behave in the exact same way, it is your job to try and help it stay on the same trajectory every run.

###### Grounding Goals and Conversation #########

Customer goals:

goals

The following is the source conversation the user simulator is trying to replicate:

{goal_convo}

####################################################

######## CURRENT (real) Conversation #####################

This is the CURRENT conversaiton the user simulator is having:

{current_convo}

Use your best judgement if the conversation is not going well, it's possible the agent is not good enough and you need to end the conversation. End the conversation by putting END_CONVERSATION after your quote.

Keep in mind the Customer goals all must be communicated in order to give the agent enough information to properly search and book.

It is critical you give consistent advice over multiple iterations of the same conversation. The best way to do that is to ground your response in the source conversation and providing quotes whenever possible.

Please write breif advice on what the user simulator should say in order to keep it consistent and aligned with the source conversation. Write this advice to the user simulatior, referring to it as "you". No yapping.:

Example:

Advice:

The user should ...

Suggested quote:

"Hello, how can I help you?"

Advice:

The conversation should be ended

Suggested quote:

"Thanks, goodbye" END_CONVERSATION

Output:

# E EFFECT OF CHANGING THE USER LLM BEHIND MULTIWOZ

We do a restricted pair of experiments ablating for the user LLM behind MultiWOZ used for evaluation at test time, to check whether the JOSH aligned models still maintain their advantage over the vanilla Llama3-8B-instruct one. We use gpt-4-turbo as the alternative user LLM.

The results are indeed positive. We find that JOSH-KTO gets average return 0.72 compared to 0.498 for the vanilla model.

