# OpenReview forum: "Sparse Rewards Can Self-Train Dialogue Agents"
_ICLR.cc/2025/Conference — Submitted to ICLR 2025_

### Official Review · Reviewer_e3Ly · 2024-10-30

**Soundness:** 2
**Presentation:** 2
**Contribution:** 2
**Rating:** 5
**Confidence:** 4

**Summary:**

This paper introduces JOSH, a self-training framework designed to enable agentic models to achieve self-alignment. The core component of JOSH is the data rollout pipeline, where an agent first interacts with a GPT-based simulator to generate multi-turn conversations that involve tool-calling responses. A critical aspect of this process is the use of beam search to create a tree-structured trajectory. From this trajectory tree, they extract SFT and preference data for subsequent fine-tuning. To evaluate JOSH, they have also curated a multi-turn tool-calling benchmark called ToolWOZ.

**Strengths:**

The proposed method is evaluated on different model backbones, even gpt-4o

The curated benchmark is useful to the community.

The presentation is mostly clear and easy to follow.

**Weaknesses:**

The primary weaknesses of this paper lie in its novelty and the experimental validation.

Novelty: The proposed framework is not particularly novel, as it builds upon concepts that have been extensively studied within the community. Techniques such as data rollouts, beam search, and supervised/preference fine-tuning have all been well-explored in prior works.

Experiments: This paper evaluates JOSH using only a single benchmark and does not provide comparisons with other robust baselines. There are numerous multi-turn tool-calling and agentic benchmarks available, such as WebLinx[1] and MINT[2]; conducting experiments on multiple benchmarks would significantly strengthen the validity of the results. Furthermore, there are several highly similar methods in this domain, such as [3,4], which should be considered as baselines to effectively demonstrate the true performance of JOSH.


[1] WebLINX: Real-World Website Navigation with Multi-Turn Dialogue
[2] MINT: Evaluating LLMs in Multi-turn Interaction with Tools and Language Feedback
[3] Trial and Error: Exploration-Based Trajectory Optimization for LLM Agents
[4] V-STaR: Training Verifiers for Self-Taught Reasoners

**Questions:**

How much does it cost to finetune 4o?

---

> ### Author Response · Authors · 2024-11-23
> **Addressing questions and concerns (R4)**
>
> We would like to thank the reviewer for the feedback. We address each of the stated weaknesses below
>
> 1. Novelty. While it's true that the individual components like data rollouts, beam search, and preference fine-tuning have been studied, our work introduces a novel combination of these elements in the context of self-training dialogue agents using sparse rewards. Specifically, JOSH leverages sparse reward simulations to autonomously generate preference-annotated data without external human feedback, which distinguishes it from existing methods that often rely on dense rewards or human annotations.
>
> In our paper, we discuss how JOSH:
>
> - Integrates sparse rewards with beam search simulations to explore and harvest optimal conversation trajectories effectively.
> - Generates both supervised and preference data from the agent's own simulations, enabling self-improvement in a way that hasn't been extensively explored.
> - Demonstrates significant performance gains across models of varying sizes, including frontier models like GPT-4, showcasing the scalability and effectiveness of our approach.
>
> We acknowledge that we could have more explicitly highlighted the novelty of our method in comparison to prior work. In the revised version, we will emphasize how JOSH differentiates itself and contributes uniquely to the field, providing a clearer articulation of its innovative aspects.
>
> 2. Experiments.
> In this paper we evaluate JOSH against **two** benchmarks, one introduced in this paper ToolWOZ (Table 2) and another external benchmark Tau-bench (Table 3). These two particular datasets were the only benchmarks used because they have critical aspects needed for JOSH to function: scenarios with multiple ground truth actions that can serve as sparse rewards. Even though the MINT[2] benchmark takes place in a tool calling mult-turn dialogue setting, it lacks a goal set of individual tool calls that can serve as sparse rewards for JOSH to function on, rather there is only a single solution that the bot is iterating towards. While it would be interesting to adapt MINT so JOSH can be applied, this task is beyond the scope of this paper. While we could adapt JOSH to perform on a web based tool using benchmark such as WebLinx[1] we aim in this paper to improve the ability of dialogue agents, thus making web navigation also out of scope for this paper.
>
> Baselines.
> JOSH introduces a novel approach to sparse reward-based alignment and self-improvement specifically tailored for multi-turn dialogue agents utilizing tools. While there are existing self-alignment methods, most are not designed for dialogue agents or do not operate effectively in multi-turn settings. To the best of our knowledge, JOSH is the first method that enables dialogue agents to self-improve their tool-calling capabilities in a multi-turn conversational context without relying on external human feedback.
>
> The approach outlined in "Trial and Error: Exploration-Based Trajectory Optimization for LLM Agents" (ETO) involves a behavioral cloning step that requires initial training on ground truth examples (human input). However, the core objective of our paper and JOSH is to "autonomously enhance LLM agent performance without external human feedback." Additionally, the ETO's contrastive training phase cannot be executed on closed-source LLMs such as GPT-4, which limits the models that can be trained with ETO.
>
> Similarly, V-STaR: Training Verifiers for Self-Taught Reasoners offers a method for advanced self-training based on output success but is designed for single-turn outputs like GSM8K or MATH. Our paper focuses on techniques for solving multi-turn dialogue problems. While adapting V-STaR for multi-step self-improvement would be interesting, it is also beyond this paper's scope. Additionally, we were not able to find a code implementation of V-STaR which is another impediment for reproducing.
>
> In our work, we provide a comprehensive baseline by experimenting with multiple training methods across various model sizes, including both small and frontier models. This thorough exploration demonstrates the scalability and effectiveness of JOSH, setting a new standard for future research in this area. Our focus on enhancing tool use within multi-turn dialogues distinguishes JOSH from other approaches and highlights its unique contribution to the field.
>
> By establishing this baseline, we aim to facilitate comparisons in future studies and encourage the development of more advanced self-alignment techniques for dialogue agents. We believe that JOSH paves the way for new possibilities in creating autonomous, efficient, and capable dialogue systems.
>
> Questions:
> The price to finetune gpt-4o is $25.000/1M training tokens and gpt-4o-mini is $3.000/1M training tokens. On average we trained from 3 million training tokens to 10 million training tokens in our experiments depending on the dataset and how many examples were used, so it cost anywhere from $75 to $250 to finetune gpt-4o.

---

> > ### Comment · Reviewer_e3Ly · 2024-11-25
> >
> > Thank you for the clarification, I will raise my score.

---

> > > ### Author Response · Authors · 2024-11-27
> > > **Update on Paper**
> > >
> > > We would like to thank you for your continued review of this paper! We have updated the paper in a number of places including addressing the comment on the cost of training gpt-4o in Lines 307-308. We have also expanded the definition of JOSH in Section 2, trimmed down Section 3 TooWOZ and provided even further analysis in Section 5 (see specific lines addressing different reviewer comments in above comments).

---

### Official Review · Reviewer_YKJK · 2024-11-04

**Soundness:** 3
**Presentation:** 3
**Contribution:** 3
**Rating:** 6
**Confidence:** 2

**Summary:**

They propose both a benchmark and a method for training multi-turn tool use dialogue agents. Their method uses beam search to find successful trajectories and uses failed paths in the beam search as negative examples. They finetune an LM on these successful/unsuccessful pairs with KTO. The benchmark is called ToolWOZ, which re-purposes the popular dialogue systems benchmark MultiWOZ to a more native LM tool use format. They evaluate their method on both ToolWOZ and another standard benchmark tau-bench. They find that their method substantially improves LLaMA 3-8B's success rate on both benchmarks. They also conduct evaluations of the robustness of each benchmark by analyzing the standard deviation of the results, finding ToolWOZ with the goal simulator to give the lowest standard deviation. Finally, they conduct some error analysis of their approach on ToolWOZ.

**Strengths:**

* They propose both a novel method and a benchmark, but they also make sure to evaluate on an existing benchmark to enable more robust comparisons.
* They conduct good analysis to demonstrate the robustness and viability of their benchmark.
* Their method demonstrates good performance gains on the tasks they study.
* The paper is overall well written and easy to follow.

**Weaknesses:**

* Their method feels a little ad-hoc. Yes, it makes sense to build off-policy preference pairs for training these models, but there are numerous ways this could be achieved and its unclear why the specific methodological decisions made in this paper are the correct ones.
* They compare to an SFT baseline, but not other RL-inspired approaches for finetuning agents, so it is unclear how well their approach compares against stronger baselines.

**Questions:**

See weaknesses.

---

> ### Author Response · Authors · 2024-11-22
> **Addressing the weaknesses pointed out (R3)**
>
> We would like to thank the reviewer for their valuable feedback, we have addressed the questions/weaknesses stated below
>
> 1. Thank you for the helpful comments! We have spent considerable time writing out comments that further define and justify the choices made in the design of JOSH. We plan on shortening section 3 ToolWOZ and incorporating these additional definitions into the final version of the paper. A list of topics that we have further fleshed out include:
>
> a. Reviewer 1, point 3 further defining JOSH’s average reward function and comparing this to other types of reward functions
>
> b. Reviewer 2, point 1 further defining what a goal set is, how we use the goal set to enforce chronological execution, and the positive implications and insights of the way we track how goals have been executed by the agent.
>
> c. Reviewer 2, point 2 explores the different ways JOSH could perform branching and why the design decisions that we made are optimal for the problem that we are trying to solve.
>
> We hope that these additional insights further illuminate why certain methodological decisions were made and how they compare to alternative approaches.
>
> 2. Thank you for this point, as you can see in the reply to Reviewer 2, point 4, we have seen good results using other RL approaches such as KTO on smaller open source models. These results can be seen in Table 2 with the metal-llama-8B-JOSH-KTO model and is further explored in the Analysis sections 5.2 and 5.3. However, for closed sourced models (GPT) we are limited to only training using supervised fine tuning through their website. Thus to experiment with JOSH on larger or frontier models, we needed to use only SFT baselines. We supplemented this by including some experiments on different branching factors and how that affected SFT training (also seen in Table 2 however we agree that larger companies should allow more exploratory training procedures and it is likely that using KTO models such as GPT would see better results.

---

> > ### Comment · Reviewer_YKJK · 2024-11-26
> >
> > Thank you for responding to my concerns! I think if you can clarify the points you mentioned in the paper, that would help it a lot. As for baselines, I was wondering how your approach would compare against, say REINFORCE or PPO. Or just generally some simple baseline that is stronger than SFT, but more straightforward and obvious than the method presented in the paper.

---

> > > ### Author Response · Authors · 2024-11-27
> > > **Updates to the paper and addressing baseline concerns**
> > >
> > > Thank you for continuing to provide insightful feedback! We have added a significant amount to the paper to address all of the concerns and further experiments that have been raised in this review process. Specifically relating to your comments about expanding our definition of JOSH, we've added a significant amount around the design choices that we've made and what some alternative approaches are:
> > > 1. a. We've updated Lines 133-142 to address how we picked our reward function and what other options there are.
> > > 1. b. We added to Lines 267-269 and Lines 156-157 to further define how goal sets work in JOSH and ToolWOZ.
> > > 1. c. We added Lines 162-171 to explore exactly why we picked branching by turn and the associated math to compare with branching by action.
> > >
> > > Thank you for your insightful point on other RL baselines such as PPO vs the use of SFT in our paper. We use SFT as a baseline in this paper rather than other online reinforcement learning based methods (PPO or REINFORCE) for three reasons. For dialogue and dialogue understanding it has been documented that some form of SFT (via offline RL example selection) does as well or better than PPO (https://arxiv.org/pdf/2307.12425). Additionally, we explore preference tuning methods over PPO due to our own computational constraints as well as our target user's. Finally, SFT is the only form of training available for closed source frontier models, and so for larger models it is not possible to experiment with other training paradigms. Thus we leave online reinforcement learning to future work.
> > >
> > > Thank you again for your continued support, we hope this is helpful explanation!

---

### Official Review · Reviewer_qkhY · 2024-11-05

**Soundness:** 2
**Presentation:** 2
**Contribution:** 2
**Rating:** 3
**Confidence:** 4

**Summary:**

This paper introduces a self-alignment approach called Juxtaposed Outcomes for Simulation Harvesting (JOSH), designed to improve dialogue agents in multi-turn, tool-calling tasks by leveraging sparse rewards. The authors propose ToolWOZ, a new simulation environment derived from MultiWOZ, for training agents to make correct API calls based on sparse reward feedback. The JOSH method aims to allow models, including smaller LLMs, to improve autonomously without relying on extensive human feedback, which is increasingly challenging to obtain as models advance.

**Strengths:**

1. The JOSH approach is a new solution for self-training dialogue agents, effectively utilizing sparse rewards to build a self-improvement feedback loop without external human evaluation.
2. By adapting MultiWOZ into ToolWOZ with a sparse reward structure, the paper provides a valuable benchmark tailored for tool-using task-oriented dialogue systems, which can benefit further research.
3. Results indicate that JOSH significantly improves models across benchmarks, demonstrating its potential as a scalable solution for optimizing agent interactions in multi-turn dialogue settings.

**Weaknesses:**

1. The concept of the "goal set" in sparse rewards is insufficiently defined, particularly how it influences the agent’s behavior and the implications of duplicating actions in a path.
2. The choice to branch at the turn level rather than the agent action level lacks a comprehensive rationale, leaving questions about its impact on computational efficiency and performance outcomes. In multiwoz dataset, the agent predicts dialogue act in each turn. The delexiclized response is then generated. The slots values are then filled in the delexiclized response to yield the final response. This process is clearly different from the one illustrated in Figure 2.
3. While considerable effort is spent on detailing ToolWOZ, the sparse reward process and its precise mechanics within JOSH are not thoroughly elaborated, reducing clarity around its contribution to the results.
4. The baseline comparisons are primarily limited to supervised fine-tuning (SFT) and variants of the sparse reward approach itself. To better contextualize the efficacy of JOSH, comparisons with other RL-based methods, particularly those designed for dialogue or tool-calling tasks, would be beneficial.

**Questions:**

As in weaknesses

**Details Of Ethics Concerns:**

N.A.

---

> ### Author Response · Authors · 2024-11-22
> **Addressing the weaknesses pointed out (R2)**
>
> 1. Thank you for pointing this out, indeed it is not clear. For the purpose of this work, our goal set per simulation is a set of APIs (or tools) that must be called with corresponding parameters. E.g., if a simulation for canceling a flight requires the ‘retrieve_reservation(confirmation_code=ABCDEF)’ API to be called with corresponding confirmation code, that is a goal and it is only achieved once called with the correct parameters. There are a number of considerations here with respect to agent behavior, goal set and the interaction with our beam search. First, our beam search is designed to follow paths once goals are hit, so this naturally will select for trajectories where goals are achieved earlier in the conversation. It is an open question whether this can be suboptimal, but changing the beam search strategy could potentially account for this. Once a goal is achieved, it is removed from the set, so the agent cannot continue to obtain rewards from making duplicate calls.  Furthermore, we force an ordering on goals to ensure they are called in the right order. This ordering is enforced by ensuring that some apis require information that can only be found by making other api calls correctly. For example, the “book_train” api call requires a train_id, which can only be found by making a correct call to “search_train”. This way, we ensure that a “book_train” api call cannot be made correctly without first searching for the correct train. Extensions to the goal set and its dynamics are an interesting topic of future work.
>
> 2. We agree that the internal action breakup of an episode [conversation] happening between an agent and a user in ToolWoz may differ significantly and not have a clear one to one correspondence to action trajectories as they used to take place in MultiWOZ.
> Casting the interaction in terms of alternating natural language token sequence generation by the agent and the user naturally requires turn level branching for ToolWoz - if one were to enforce a strict agent action based framework that decomposes the notion of a turn, each token generation step by the agent would have to be considered an “action” ; this would both lead to unnaturally long “action” sequences and make the intuitive passing of control between agent and user a lower-frequency intermediate event that happens once every k actions, rather than being in natural lockstep with the granularity. Also, the user here is a part of the environment, so the user turn that follows an agent turn can be seen as a natural part of the environment’s state transition function. With ToolWoz , we were looking for an approximate example level correspondence to MultiWOZ in terms of the initial information as well as the goals, rather than exactly creating a one to one mappable replication of the action dynamics or trajectories that would take place in MultiWOZ.
>
> Binary trees have a number of 2^h-1 leaf nodes where h is the height of the tree, since JOSH splits at the turn level we can expect t=log_2(max_branches)+1 to be the number of turns t before JOSH can no longer expand the tree. There are roughly 3 actions a per turn on average, so t=3a and thus the number of turns allowed before branching would stop when action splitting is t = (log_2(max_branches)+1)/3. Thus when max_branchs=8 which is used throughout the paper to keep costs reasonable (around $100) we could perform either t=4 turns while still splitting, or t=4/3 turns when splitting on actions. While splitting on actions may provide more diversity, over the course of a multi turn dialogue we can explore more possible paths deep in the tree for the same max_branches when splitting on turns.
>
> In reference to Figure 2, thank you for pointing this out and we have revised the figure to reflect the correct system design and will revise this figure in the final version of the paper.
>
> 3. We plan to compress section 3 ToolWOZ in order to spend more time rigorously defining the JOSH process defined in both these replies and replies to the other reviewers into the final version of the paper. For a deep dive into the reward process, we have further outlined many details in points 1 and 2 of this comment, as well as the comment to Reviewer 1.
>
> 4. Thank you for this point and we agree wholeheartedly that other RL based methods should be explored further using this approach. For open source models we include the exploration of other RL approaches (metal-llama-8B-JOSH-KTO, in Table 2) and find better results than variants of supervised fine tuning. We also further explore the benefits of using KTO in the Analysis sections 5.2 and 5.3. However, for closed sourced models (GPT) we are limited to only training using supervised fine tuning through their website. Ideally, companies such as OpenAI would allow for more exploratory training techniques but in order to show that JOSH works for all sized models and frontier models we chose to take advantage of the limited training that was available to us.

---

> > ### Comment · Reviewer_qkhY · 2024-11-26
> >
> > Thanks for your response. However, after reading the response, I feel that the work will need further modification before publication. Even earlier days methods did not 'force an ordering on goals' on the MultiWOZ dataset for strategy planning or response generation. Such process would largely affect the generality of the method to other datasets or settings. Also, it feels that this work needs further improvements via comparing with better baselines.

---

> > > ### Author Response · Authors · 2024-11-27
> > > **Clarification on Goal Ordering and Paper Update**
> > >
> > > We would like to thank the reviewer for thoughtfully considering our comment!
> > >
> > > 1. We would like to provide some further clarification on the subject of goal ordering in ToolWOZ and JOSH in general. JOSH does not require any ordering for the goals in it's goal set. JOSH is built to be flexible and any constraints that one wants to enforce (or not) could be implemented. In reference to ToolWOZ, goal api calls are not strictly ordered either. Rather, for some "booking" api calls, some of the information needed for the booking must be found by using a "search" api call to gather information. This is similar to many real world scenarios where the agent has imperfect information and must use tools to gather more information. However, the reward that an agent gets from JOSH when executing a ToolWOZ tool call that is still in the goal set still has no ordering and is counted regardless of what other tool calls have been executed thus far. We have not fundamentally changed the task in this way from any prior works.
> > > We apologize for any confusion our above comment cause in terms of this matter. We have updated the paper to also address this point in Lines 267-269 and Lines 156-157.
> > >
> > > 4. Thank you for your insightful point on other RL baselines such as PPO vs the use of SFT in our paper. We use SFT as a baseline in this paper rather than other online reinforcement learning based methods (PPO or REINFORCE) for three reasons. For dialogue and dialogue understanding it has been documented that some form of SFT (via offline RL example selection) does as well or better than PPO (https://arxiv.org/pdf/2307.12425). Additionally, we explore preference tuning methods over PPO due to our own computational constraints as well as our target user's. Finally, SFT is the only form of training available for closed source frontier models, and so for larger models it is not possible to experiment with other training paradigms. Thus we leave online reinforcement learning to future work.
> > >
> > > Additionally we have updated the paper to address your other comments
> > > 2. Figure 2 has been updated to better reflect the process of a MultiWOZ style agent. We have also updated Lines 162-171 to reflect the math and design decisions made with respect to branching on actions vs turns.
> > > 3. Section 3 has been significantly compressed to add more room for further defining JOSH in section 2 and additional analysis in section 5.

---

### Official Review · Reviewer_dqjx · 2024-11-09

**Soundness:** 2
**Presentation:** 3
**Contribution:** 2
**Rating:** 5
**Confidence:** 3

**Summary:**

The paper introduces JOSH (Juxtaposed Outcomes for Simulation Harvesting), a self-alignment framework for large language model (LLM) agents to enhance multi-turn dialogue capabilities without human feedback, addressing the impracticality of traditional feedback-driven methods. JOSH leverages sparse reward signals within simulated dialogues to allow the model to self-improve, specifically targeting multi-turn tool-calling skills in task-oriented dialogues. The authors also introduce ToolWOZ, a dataset and benchmark based on MultiWOZ 2.0, designed to evaluate tool-usage in dialogue settings. Experimental results demonstrate that a fine-tuned LLaMA-3B model exhibits a 74% increase in Success Rate, and gpt-4o also shows improvements following JOSH self-alignment. Additional experiments on other public benchmarks indicate that JOSH does not degrade the model’s general performance.

**Strengths:**

* This paper presents a novel approach to self-alignment in dialogue agents using sparse rewards, reducing reliance on costly human feedback.
* ToolWOZ fills a gap in existing evaluation frameworks by focusing on tool usage in multi-turn dialogue settings, adapting MultiWOZ to emphasize real-world API interactions.
* JOSH demonstrates significant improvements in success rates and tool-call accuracy, particularly for smaller models, validating its effectiveness.

**Weaknesses:**

* The paper does not assess how well the user simulator aligns with real human interactions.
* The evaluation of API calls lacks depth, as it does not separate analyses of API names and parameters.
* The design of the average reward function is not thoroughly examined, missing a discussion of alternative reward structures and their potential effects on agent behavior.
* The related work section does not cover relevant advancements in language agents for multi-turn dialogues.

**Questions:**

* How does JOSH compare to other sparse reward-based alignment or self-improvement approaches?
* Could strategies like reflection, which are often beneficial for tree-search and multi-turn tasks, enhance JOSH’s effectiveness if integrated?

---

> ### Author Response · Authors · 2024-11-22
> **Addressing questions and weaknesses 1 and 2 stated (R1)**
>
> q1. JOSH provides a baseline for sparse reward based self-improvement in multi-turn dialogue agents. Particularly when considering the improvement in tool calling capabilities, we provide experiments using multiple types of training and along various sized models in order to thoroughly explore this baseline approach so that others may compare to it in future works. Other self-alignment approaches do exist for dialogue agents, however most notable approaches are not built for a multi-turn setting nor for dialogue. We also target improvement of tool use in this multi-turn dialogue setting, which further distinguishes us from any other baseline approaches.
>
> q2. We provide experiments with reflection using ReACT based prompts in both ToolWOZ and TauBench results, across all model sizes. We do find that reflection techniques enhances JOSH's effectiveness as all ReACT based models outperform themselves when trained on JOSH data.
>
> 1. We perform an additional analysis comparing the goal based user simulator to the ground truth human conversations in MultiWOZ. We evaluate across three dimensions (naturalness, conciseness, and redundancy) using prompts from the paper LLM-RUBRIC: A Multidimensional, Calibrated Approach to Automated Evaluation of Natural Language Texts. The prompts evaluate the user messages in an entire conversation, assigning a score 1-4 where 4 is the best. We take the average over all 450 conversations in the ToolWOZ test set. We use Claude Sonnet 3.5 as the evaluator. The results are as follows:
>
> dimension   human   bot
>
> naturalness   4            4
>
> conciseness  3.98      3.94
>
> redundant     3.59      3.42
>
> As we see, both humans and the user simulator are scored as very natural. The conciseness of the user simulator is slightly worse than the human score, which we attribute to the tendency for the user simulator to be verbose in its replies. Finally, the redundancy score for the user simulator is worse than a human, but still achieves a score of 3.42. Our analysis shows that this drop is due to agent errors where information is re-requested, and the user simulator is more willing to reiterate information where humans are less likely to repeat critical information.
>
> 2. We have performed a deeper analysis of the API calls and where errors are arising with different models and we plan to include this in the final version of our paper. The table below shows the amount of failed api calls split by type of api. Notably, the search_train and search_attraction apis still have a large gap when comparing sft and kto trained models, where sft is far more likely to fail.
>
>         Method book_train  search_hotel  search_train  book_hotel  search_attraction  book_restaurant  search_restaurant
>         base	83	77	84	59	96	49	47
>         sft	56	44	81	36	77	32	36
>         kto	45	49	50	41	40	27	31
>
> To further investigate this phenomenon, we measured the frequency of required argument groups for search_train and search_attraction that sft failed to call. We observe that while search_train failed calls with “leaveAt” argument steadily decrease from base to sft, calls with the “arriveBy” argument actually slightly increase in failures with sft from base. This pattern is not consistent with kto training, however, where the failures in both sections decrease significantly from base. We find that this phenomenon is due to sft training commonly leaving out arguments when writing API calls, and in the case of “arriveBy” api calls, the “departure” parameter is commonly left out. KTO however avoids this pitfall by training on apis with too few parameters as negative examples, and is thus far more likely to include all parameters.
>
> search_train failure
>
> (base) [(['day', 'departure', 'destination', 'leaveAt'], 37), (['arriveBy', 'day', 'departure', 'destination'], 47)]
>
> (sft) [(['day', 'departure', 'destination', 'leaveAt'], 31), (['arriveBy', 'day', 'departure', 'destination'], 50)]
>
> (kto) [(['day', 'departure', 'destination', 'leaveAt'], 22), (['arriveBy', 'day', 'departure', 'destination'], 28)]
>
> We observe a similar phenomenon in the search_attraction api, where arguments including “area” almost never drop. We observe that this is due to two reasons. First, the sft model often neglected to use the “type” argument alongside the “area” argument, choosing to often only fill in the “area”. Also, the “area” argument was commonly being filled in as “all” for many sft conversations even though this is not a valid value for the area parameter. The KTO trained model manages to avoid many of these pitfalls as well as these invalid apis are commonly found in the negative examples.
>
> search_attraction failure
>
> (base) [(['area'], 9), (['area', 'type'], 32), (['name'], 29), (['type'], 26)]
>
> (sft) [(['area'], 8), (['area', 'type'], 30), (['name'], 21), (['type'], 18)]
>
> (kto) [(['area'], 2), (['area', 'type'], 15), (['name'], 12), (['type'], 11)]
>
> We plan to expand the error analysis in section 5.2 with these findings.

---

> > ### Author Response · Authors · 2024-11-22
> > **Addressing weaknesses 3 and 4 stated (R1)**
> >
> > 3. **Design of the Average Reward Function**
> >
> > We chose the average reward function to balance efficiency and effectiveness in multi-turn dialogues. By averaging rewards over the total number of goals, the agent is incentivized to accomplish all objectives while minimizing the number of API calls and dialogue turns. This approach discourages unnecessary actions and promotes concise, goal-oriented behavior.
> >
> > **Alternative Reward Structures and Their Implications**
> >
> > We considered several alternative reward structures:
> > - **Cumulative Reward**: This approach sums all rewards without normalization. While straightforward, it may encourage the agent to make excessive API calls to maximize the total reward, leading to inefficient interactions. Our goal is to have the agent resolve customer issues with the minimal necessary API calls, so cumulative rewards are less suitable.
> > - **Per-Turn Reward**: Assigning rewards at each turn provides dense feedback, potentially accelerating learning. However, it requires per-turn level annotations, which are expensive to obtain. Although leveraging an LLM as a judge to approximate per-turn rewards is possible, it demands significant resources to develop effectively. We leave this exploration for future work.
> > - **Sparse Goal-Based Reward**: Similar to our method, this rewards the agent only upon achieving specific goals. The key difference is that traditional sparse rewards grant a single reward at the end of the conversation upon completing all goals. In contrast, our average reward function provides partial rewards as each goal (API call) is achieved during the conversation. This offers earlier feedback, helping the agent adjust its behavior in real-time.
> > - **Shaped Reward**: Incorporating intermediate rewards can guide the agent toward goals more effectively. However, designing appropriate shaping rewards is complex and may require an LLM judge to evaluate intermediate actions, adding to the development time and resource requirements. We consider this an area for future investigation.
> >
> > **Potential Effects on Agent Behavior**
> > - **Cumulative Reward**: May encourage inefficient behavior by incentivizing the agent to perform unnecessary actions to accumulate more rewards, leading to longer and less efficient dialogues.
> > - **Per-Turn Reward**: Could cause the agent to prioritize immediate, potentially low-value actions that yield instant rewards, detracting from achieving the overall conversation goals.
> > - **Sparse Goal-Based Reward (Our Approach)**: By providing partial rewards for each achieved goal, the agent is motivated to focus on completing all tasks efficiently, enhancing both effectiveness and dialogue conciseness.
> > - **Shaped Reward**: While potentially improving learning speed, it risks overcomplicating the reward structure and may inadvertently encourage the agent to optimize for intermediate rewards rather than the final objectives.
> >
> > By adopting the average reward function with partial sparse rewards, we effectively promote efficient goal completion without the complexities and potential drawbacks of alternative reward structures. This design choice aligns with our objectives of fostering efficient, goal-oriented dialogues while maintaining a straightforward and effective reward mechanism.
> >
> > The related work section does not cover relevant advancements in language agents for multi-turn dialogues.
> >
> > 4. We are adding a section to the final paper relating to the strides multi turn dialogue has taken. We will again reference the advent of LLMs as stated in the introduction, and also reference the advent of tool use in agents (Gorilla LLM, toolbench), and approaches that have been used to improve the function of LLM Agents such as ReACT and Chain-of-thought.

---

> > > ### Comment · Reviewer_dqjx · 2024-11-26
> > > **Official Comment by Reviewer dqjx**
> > >
> > > Thank you for your detailed response and the additional experiments addressing questions 1 and 2. While I appreciate the effort put into these analyses and the clarifications provided, I believe that the additional experiments and findings should be carefully expanded and integrated into the paper before publication. Therefore, I will keep my score.

---

> ### Author Response · Authors · 2024-11-27
> **Expanding the comments and experiments into the paper**
>
> You are absolutely correct, and so we have just revised the PDF version to address all of the comments from reviewers! In reference to your comments, we have updated the paper in the following places:
> 1. Lines 410-418 to provide analysis relating to the goal based user simulator compared to ground truth humans
> 2. Lines 432-444 to provide additional analysis of API errors
> 3. Lines 133-142 defining why we chose our average reward functions and showing other options
> 4. Lines 486-494 adding relevant advancements in language agents for multi-turn dialogues into the related works.
>
> We would like to thank the reviewer for their ongoing efforts!

---

### Author Response · Authors · 2024-12-03
**Final Paper Draft Comments**

Thank you to all of the reviewers for your insightful feedback and ongoing support. We've revised our paper to incorporate your suggestions and enhance its quality. Let us know if you have any further comments ahead of the comment deadline today.

---

### Meta-Review · Area_Chair_8HRo · 2024-12-17

**Metareview:**

This work proposes an LLM self-training approach based on sparse reward simulation. The authors also plan to release their code and data. The reviewers appreciate the general idea of this work, the value of the ToolWOZ benchmark to the community, the performance improvements achieved, and the clarity in presentation. Concerns are also raised, however, primarily regarding experimental validation, the design of the simulator (validity, reward function and API call design) as well as concerns around novelty and relation to existing works. The authors provide extensive and detailed responses but the reviewers are not convinced.

**Additional Comments On Reviewer Discussion:**

The discussions mostly focus on clarifying design choices around JOSH. The authors agree that more comparisons to stronger baselines are needed but also emphasize the positive results already in the paper. The authors also conducted a deeper analysis on the API calls issue and present new results (that, however do not seem to convince the reviewer). Overall, while many of the concerns seem to be clarifications, some are more fundamental (e.g. comparing against stronger baselines) and therefore I believe this work is not yet ready for publication.

---

### Decision · Program_Chairs · 2025-01-22

Reject